# Object-Centric Dexterous Manipulation from Human Motion Data

**Yuanpei Chen**[1,2]**, Chen Wang**[1]**, Yaodong Yang**[2]**, C. Karen Liu**[1]

[1]Stanford University, [2]Peking University

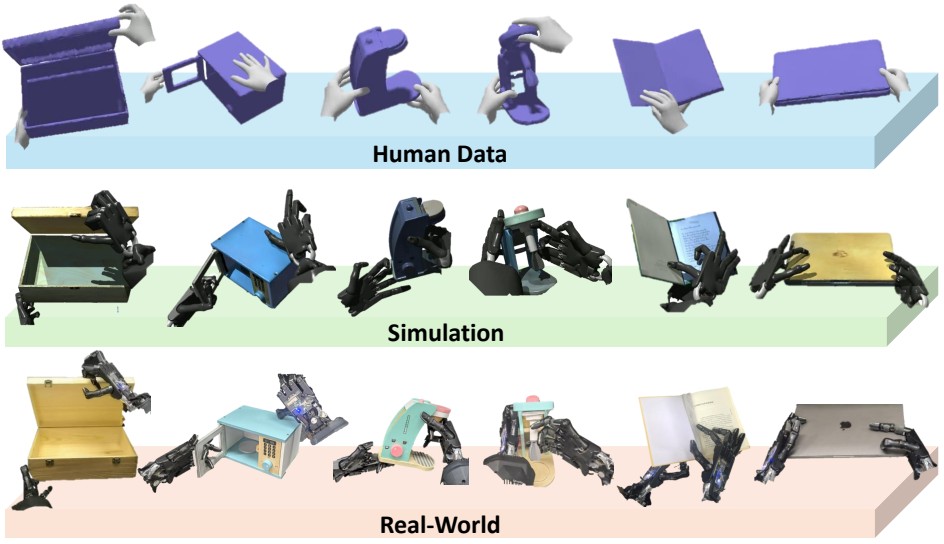

Figure 1: Our system uses human hand motion capture data and deep reinforcement learning to train dexterous robot hands for effective object-centric manipulation (i.e., learning to manipulate an object to follow a goal trajectory) in both simulation and real world.

**Abstract:** Manipulating objects to achieve desired goal states is a basic but important skill for dexterous manipulation. Human hand motions demonstrate proficient manipulation capability, providing valuable data for training robots with multi-finger hands. Despite this potential, substantial challenges arise due to the embodiment gap between human and robot hands. In this work, we introduce a hierarchical policy learning framework that uses human hand motion data for training object-centric dexterous robot manipulation. At the core of our method is a high-level trajectory generative model, learned with a large-scale human hand motion capture dataset, to synthesize human-like wrist motions conditioned on the desired object goal states. Guided by the generated wrist motions, deep reinforcement learning is further used to train a low-level finger controller that is grounded in the robot's embodiment to physically interact with the object to achieve the goal. Through extensive evaluation across 10 household objects, our approach not only demonstrates superior performance but also showcases generalization capability to novel object geometries and goal states. Furthermore, we transfer the learned policies from simulation to a real-world bimanual dexterous robot system, further demonstrating its applicability in real-world scenarios. Project website: `https://cypypccpy.github.io/obj-dex.github.io/`.

**Keywords:** Dexterous Manipulation, RL, Learning from Human

---

Correspondence to Chen Wang <chenwj@stanford.edu>

8th Conference on Robot Learning (CoRL 2024), Munich, Germany.

# 1 Introduction

Developing bimanual multi-fingered robotic systems capable of handling complex manipulation tasks with human-level dexterity has been a longstanding goal in robotics research. When a robot engages general manipulation tasks beyond a simple pick-and-place, the definition of a *task* can be difficult to establish. Previous works defined manipulation tasks in a variety of ways, including the state of the environment [1, 2], symbolic representations [3, 4], or language descriptions [5, 6]. Regardless of how the goals are specified, a common element across these definitions is an object-centric perspective focusing on the state of the objects being manipulated. As such, the goal of our work is to train a policy for a bimanual dexterous robot to manipulate the objects according to the task goal defined as a sequence of object pose trajectories.

Prior works primarily utilize deep reinforcement learning (RL) to learn object-centric dexterous manipulation skills [7–9]. Despite the success of these methods in tasks such as in-hand object reorientation [10–12], they typically focus solely on learning the movements of the fingers, neglecting the integrated coordination of both arms and hands. Training RL policy that controls both robot arms and two multi-finger hands is possible in theory, but presents substantial challenges in practice due to the high degree of freedom of the robot action space. Imitation learning (IL) can potentially tackle this challenge by leveraging the guidance from human motion data to assist policy learning. However, another challenge arises due to the morphological differences between human and robotic hands, often referred to as the "embodiment gap". For instance, some robotic hands are designed with only four fingers, each significantly larger than a human's, making it difficult to retarget the human hand trajectories to the robot hand while achieving the intended manipulation tasks.

One critical observation is that human finger motions are not consistently useful across various manipulation tasks due to the embodiment gap. In contrast, human wrist motions offer valuable information less sensitive to the embodiment gap, such as *where* to place the palm and *how* to interact with the objects in 3D space. Such motion cues significantly reduce the complexity of the high-dimensional action space in RL training, allowing it to focus on exploring finger motions to achieve the object-centric task goal. Based on this observation, we propose a hierarchical policy learning framework consisting of a high-level planner for the wrist and a low-level controller for the hand. The high-level planner is a generative-based policy, trained by imitation learning with human wrist movements, to generate robot arm actions conditioned on a desired trajectory of the object's movements. Based on the generated arm motions, the low-level controller outputs fine-grained finger actions learned through RL exploration rather than imitation of human data. The reward function for the RL training is the likelihood between the object's movements during interaction and the reference trajectory. By harnessing the strengths of both RL and IL, we enable the robot to adapt to its own hand embodiment while keeping the training tractable by refining the action space using human data.

To ensure that the learned policy can adapt to a variety of reference object trajectories, not just a single scenario, we utilize ARCTIC [13], a comprehensive dataset with 51 hours of diverse hand-object manipulation motion capture sequences. Our experiments demonstrate that the learned policy exhibits generalization to novel object geometries and unseen motion trajectories. In addition, we successfully transfer our policy from simulation environments to a real-world bimanual dexterous robot, further validating its practical applicability in real-world manipulation tasks.

# 2 Related Works

## 2.1 Dexterous Manipulation

Dexterous manipulation is a long-standing research topic in robotics [14–17]. Traditional methods rely on analytical dynamic models for trajectory optimization [14, 15, 17, 18], which fall short in complex tasks due to the simplification of contact dynamics. [19] and [20] propose model-based optimization methods to solve the contact-rich dexterous manipulation task, but a correct dynamics model of the environment is required. Recently, deep reinforcement learning (RL) has showcased promising results in training dexterous manipulation skills such as in-hand object reorientation [10–12, 21–28], bimanual manipulation [7, 8, 29], sequential manipulation [30–32], and human-like activities [33]. [24] learn a grasping policy to make the object follows a manual designed trajectory, while we focus on more

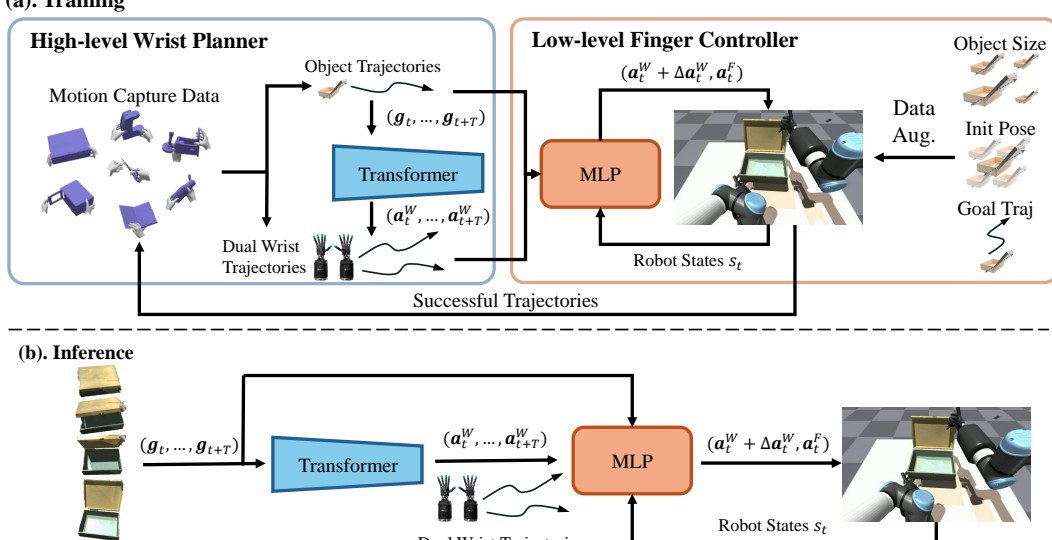

Figure 2: Overview of our framework. (A) Training: Firstly, we use human motion capture data to train a generation model to synthesize dual hand trajectory conditions on object trajectory. Then we use the RL to train a low-level robot controller conditioned on the dual hand trajectory generated by the trained high-level planner. During this process we augment the data in simulation to improve the high-level planner and low-level controller simultaneously (B) Inference: Given a single object goal trajectory, our framework generates dual hand reference trajectory and guides the low-level controller to accomplish the task.

challenging tasks. Despite the progress, successfully training a dexterous RL policy often requires extensive reward engineering and system design, which limits its practicality in some scenarios. Besides RL, imitation learning (IL) is also widely used for training dexterous policies [34, 35]. By performing supervised-learning with human teleoperation data [36–40], prior works show impressive results in dexterous grasping [41, 42] and general manipulation tasks [43–50]. However, teleoperation data are often expensive to collect due to the requirement of a real-world robot system.

## 2.2 Learning from Human Motion

Recently, learning from human motion data has started to receive more attention because it allows scaling up data collection without robot hardware. Prior works leverage human videos [51–56], motion capture data [57–62] to extract valuable motion hints for manipulation such as trajectory-level plans [51, 59], object affordance [53] and motion priors [52, 55]. For dexterous manipulation, VideoDex [55], DexMV [43], DexTransfer [58] and DexCap [63] showcase the potential of using analytical methods (e.g., inverse kinematics) to retarget human hand motion to robot hardware, such as matching joint angles [43, 55], fingertip positions [63]. [64] showing independent control of the fingers and arm for capturing detailed hand manipulation of objects. DexPilot [38] formulate the retargeting objective as a non-linear optimization problem and retarget human motion to robot by minimizing a cost function. However, due to the embodiment gap between human and robot hands, position-based retargeting methods do not guarantee the replication of task success. In contrast, our approach uses human data as guidance for RL training, which learns the motion retargeting conditioned on the robot's embodiment. Notably, [34, 65–69] share the same idea of utilizing human data as guidance or reward for reinforcement learning. While these works focus on learning human whole-body control in simulation, we study dexterous manipulation with multi-fingered robotic hands and transfer the learned policy from simulation to the real world.

## 3 Task Formulation

The goal of an object-centric manipulation task is to let the robot physically interact with the object to achieve the desired motion trajectory. We define the motion trajectory as the sequence of the object's

$SE(3)$ transformation $G = (\boldsymbol{g}_1, \boldsymbol{g}_2, ..., \boldsymbol{g}_T)$, where each time step $\boldsymbol{g}_i = (\boldsymbol{g}_i^R, \boldsymbol{g}_i^T, g_i^J)$ consists a 3D rotation $\boldsymbol{g}_i^R$, a 3D translation $\boldsymbol{g}_i^T$, and the joint angle $g_i^J$. $g_i^J$ can be omitted if the object is a single rigid body. We then formulate an object-centric manipulation task as a Markov Decision Process (MDP) $\mathcal{M} = (\boldsymbol{S}, \boldsymbol{A}, \pi, \mathcal{T}, R, \gamma, \rho, G)$, where $\boldsymbol{S}$ is the state space, $\boldsymbol{A}$ is the action space, $\pi$ is the agent's policy, $\mathcal{T}(\boldsymbol{s}_{t+1}|\boldsymbol{s}_t, \boldsymbol{a}_t)$ is the transition distribution, $R$ is the reward function, $\gamma$ is the discount factor, and $\rho$ is the initial state distribution. The policy $\pi$ conditions on the reference object state trajectory $G$ and the current state $\boldsymbol{s}_t$, and generates robot action distributions $\boldsymbol{a}_t$ to maximize the likelihood between the future object states $(\boldsymbol{s}_{t+1}, \boldsymbol{s}_{t+2}, ..., \boldsymbol{s}_{t+T})$ and the reference trajectory $G$. This formulation is versatile to adapt to downstream tasks, such as moving objects to desired locations and in-hand object re-orientation with bi-manual hands.

## 4 Method

In this section, we introduce our framework for object-centric manipulation. The overview of the framework is shown in Figure 2. Our framework consists of three parts: high-level planner (Section 4.1), low-level controller (Section 4.2) and the data augmentation loop (Section 4.3). The details of our sim-to-real policy transfer are introduced in Section 4.4.

### 4.1 High-Level Planner

One of the key challenges in training a dexterous robot policy, especially with bi-manual dexterous hands, is managing the high-dimensional action space. Despite the embodiment gap between human and robot hands, human hand motion, particularly wrist movements, provides highly informative hints for how to interact with objects and environments. Such motion hints not only narrow the action space for the robot but also provide guidance toward the task goal. Based on this observation, we first train a generative model to synthesize wrist motion by imitating human wrist movements in the motion capture data.

We train a Transformer-based generative model $\pi^H$ that takes object category ID $c$, and the desired object motion trajectory $G = (\boldsymbol{g}_t, \boldsymbol{g}_{t+1}, ..., \boldsymbol{g}_{t+T})$ as inputs and outputs a sequence of 6-DoF wrist actions $(\boldsymbol{a}_t^W, \boldsymbol{a}_{t+1}^W, ..., \boldsymbol{a}_{t+T}^W)$, where each action $\boldsymbol{a}_i^W = (\boldsymbol{p}_i^l, \boldsymbol{p}_i^r)$ consists the 6-DoF pose of the left hand $\boldsymbol{p}_i^l$ and right hand $\boldsymbol{p}_i^r$ in SE(3). In our experiments, we use $T = 10$. We leverage the entire ARCTIC dataset [13] to train $\pi^H$ with a behavior cloning algorithm. The total training samples consist of 339 sequences of human hand mocap trajectories, totaling 2.1 million steps.

### 4.2 Low-Level Controller

Based on the generated wrist trajectory from the high-level planner $\pi^H$, the low-level controller $\pi^L$ can start from a reasonable wrist pose and focus on learning the fine-grained finger motions to physically interact with the object to achieve the task goal $G$. We use Proximal Policy Optimization (PPO) [70] to train $\pi^L$. The policy $\pi^L$ takes the current observation $\boldsymbol{s}_i$, the desired object motion trajectory $G = (\boldsymbol{g}_t, \boldsymbol{g}_{t+1}, ..., \boldsymbol{g}_{t+T})$, and a sequence of 6-DoF wrist actions $(\boldsymbol{a}_t^W, \boldsymbol{a}_{t+1}^W, ..., \boldsymbol{a}_{t+T}^W)$ generated by high-level planner as inputs, and outputs the finger joint action $\boldsymbol{a}_t^F$. Here the observation $\boldsymbol{s}_t$ contains the object pose and robot proprioception. The reward function is defined as $r_t = \exp^{-(\lambda_1 * \|g_t^R - \hat{g}_t^R\|_2 + \lambda_2 * \|g_t^T - \hat{g}_t^T\|_2 + \lambda_3 * \|g_t^J - \hat{g}_t^J\|_2)}$, aiming to minimize the distance between object's movements and the desired goal trajectory. $\hat{g}_t^R$, $\hat{g}_t^T$ and $\hat{g}_t^J$ is the current 3D translation, 3D rotation and joint angle of the object respectively. $\lambda_1$, $\lambda_2$, and $\lambda_3$ are the hyperparameters to balance the weight of each component of the reward. In some scenarios, when the robot's hand is larger than the human's, the generated wrist actions from the high-level planner $\pi^H$ need to be adjusted accordingly. To achieve this, $\pi^L$ learns to output a residual wrist action $\Delta \boldsymbol{a}_t^W$ within a fixed range ($\pm 4$ centimeters for transition and $\pm 0.5$ radian for rotation). With the residual wrist actions, the policy can now adjust the wrist position to better synthesize the motion of the robot hand. The final robot action is a combination of $(\boldsymbol{a}_t^W + \Delta \boldsymbol{a}_t^W, \boldsymbol{a}_t^F)$. Please refer to Appendix B for more detail about the observation space and the reward function.

### 4.3 Data Augmentation for Generalization

Training policy only with the ARCTIC dataset [13] limits the diversity of task objects and generated motions. For instance, given a box twice the size of the one used in the dataset, it is challenging for the

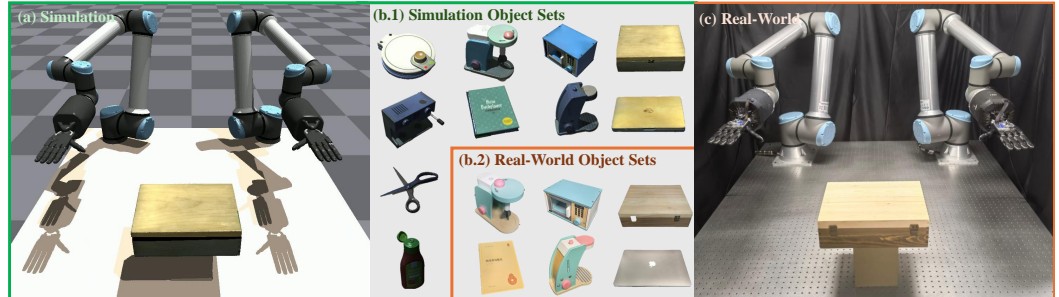

Figure 3: Overview of the environment setups. (a) Workspace of the simulation. We employ two Shadow Hands, each individually mounted on separate UR10e robots, arranged in an abreast configuration. (b.1) Object sets in the simulation. (b.2) Object sets in the real-world. (c) Workspace of the real-world, mirroring the simulation, the robot system uses the same Shadow Hands and UR10e robots as the simulation.

learned policies to manipulate the box effectively because of the unseen object geometry. To improve the generalization capability, it is critical to augment the data and train the policies to generalize in these scenarios. Thereby, we propose Data Augmentation Loop (DAL) to handle different object geometries and goal trajectories during training $\pi^L$.

Specifically, we introduce three types of augmentation during the RL training of the $\pi^L$: randomizing the object's size in all three dimensions (width, length, height), randomizing the object's initial pose, and modifying the goal trajectories of the object with waypoint interpolation. The low-level controller then learns to adapt to the augmented scenarios during the RL training. Finally, we collect the successful motion trajectories executed by the low-level controller and add them to the training dataset to fine-tune the high-level planner. By leveraging the adaptability of the RL training process, we can synthesize novel motion trajectories to improve the policy's generalization beyond the scope of the original training data. Detailed implementations of the Data Augmentation Loop can be found in Appendix A.

## 4.4 Sim-to-Real Transfer

When deploying the policy to the real world, some observation cannot be accurately estimated, such as joint velocity and object velocity. We use the teacher-student policy distillation framework [11, 21, 71] to remove the dependence on these observation inputs from the policy. In real-world deployments, our system uses four top-down cameras to perform articulated object pose estimation. We use FoundationPose [72] to estimate the 6-DoF pose of different parts of the articulated object and calculate the joint angle between them. The entire object pose tracking system runs at the speed of 15Hz. The learned policies generate actions to control the robot, and the low-level controller is processed by a low-pass filter with an exponential moving average (EMA) smoothing factor [11] to reduce the jittering motions of the robot fingers. For more details on the sim-to-real setups, please refer to Appendix C.

## 5 Experiments

The experiments are designed to answer the following research questions: (1) Can the high-level planner generalize to unseen trajectories and unseen objects? (Sec. 5.1) (2) Does our hierarchical approach help bridge the embodiment gap between human and robot hands? (Sec. 5.2) (3) Can our trained policy generalize to unseen object geometries and goal trajectories? (Sec. 5.3) (4) Can we transfer the policy from simulation to a real-world bimanual dexterous robot system? (Sec. 5.4). In this section, we first introduce our training data and baseline setups, followed by the results for each research question. The experimental setups are shown in Figure.3. Our simulation experiments are all evaluated in 10 different seeds, and the real-world experiments are all evaluated in 20 different trials. The small numbers in each table represent the standard deviations across different seeds and trials.

**Data.** The ARCTIC dataset consists of 10 objects, each with 20 sequences of motion capture data. We choose 9 objects, each with 16 sequences of data, as our training set. We left the remaining objects and motion sequences as the unseen testing set.

| | | MLP | RNN | Ours |
|---|---|---|---|---|
| Trained | TE | $5.4_{\pm0.2}$ | $5.0_{\pm0.1}$ | $\mathbf{4.2}_{\pm0.2}$ |
| | OE | $14.6_{\pm0.7}$ | $12.4_{\pm0.8}$ | $\mathbf{9.6}_{\pm0.6}$ |
| Unseen Traj | TE | $5.6_{\pm0.5}$ | $6.2_{\pm0.3}$ | $\mathbf{5.0}_{\pm0.8}$ |
| | OE | $9.4_{\pm1.0}$ | $9.2_{\pm0.5}$ | $\mathbf{7.2}_{\pm0.7}$ |
| Unseen Object | TE | $18.2_{\pm1.5}$ | $17.7_{\pm1.2}$ | $\mathbf{12.4}_{\pm1.1}$ |
| | OE | $109.4_{\pm5.8}$ | $82.2_{\pm4.2}$ | $\mathbf{75.5}_{\pm4.7}$ |

Table 1: Results for the high-level planner

| | Fingertip Mapping | Vanilla RL | Ours |
|---|---|---|---|
| Box | $13.1_{\pm1.8}$ | $20.4_{\pm1.8}$ | $\mathbf{69.8}_{\pm6.6}$ |
| Micro. | $60.6_{\pm2.7}$ | $56.1_{\pm9.8}$ | $\mathbf{100}_{\pm0.0}$ |
| Laptop | $9.2_{\pm0.5}$ | $8.6_{\pm1.4}$ | $\mathbf{76.7}_{\pm4.1}$ |
| Coffee. | $8.2_{\pm0.6}$ | $9.2_{\pm3.1}$ | $\mathbf{74.8}_{\pm3.8}$ |
| Mixer | $8.3_{\pm2.1}$ | $10.8_{\pm3.2}$ | $\mathbf{82.8}_{\pm2.1}$ |
| Notebook | $4.5_{\pm0.1}$ | $4.5_{\pm0.3}$ | $\mathbf{64.3}_{\pm8.4}$ |

Table 2: Results for the real-world experiments

| | Fingertip Mapping | Finger Joint Mapping | Vanilla RL | Ours (w. FR) | Ours (w.o. DAL) | Ours |
|---|---|---|---|---|---|---|
| Box | $14.6_{\pm0.3}$ | $8.9_{\pm0.2}$ | $23.5_{\pm3.5}$ | $56.2_{\pm7.4}$ | $100_{\pm0.0}$ | $\mathbf{100}_{\pm0.0}$ |
| Coffee Maker | $9.3_{\pm0.6}$ | $9.0_{\pm0.5}$ | $10.7_{\pm2.7}$ | $78.6_{\pm1.6}$ | $71.5_{\pm2.6}$ | $\mathbf{86.1}_{\pm5.5}$ |
| Espresso Machine | $22.2_{\pm0.4}$ | $7.0_{\pm0.8}$ | $14.3_{\pm1.5}$ | $70.7_{\pm3.5}$ | $75.4_{\pm4.3}$ | $\mathbf{81.1}_{\pm8.6}$ |
| Ketchup | $14.8_{\pm0.7}$ | $9.5_{\pm0.2}$ | $4.9_{\pm2.7}$ | $15.2_{\pm1.7}$ | $21.8_{\pm7.2}$ | $\mathbf{41.2}_{\pm13.3}$ |
| Microwave | $38.7_{\pm0.2}$ | $27.5_{\pm0.6}$ | $43.5_{\pm2.4}$ | $61.2_{\pm5.3}$ | $100_{\pm0.0}$ | $\mathbf{100}_{\pm0.0}$ |
| Mixer | $21.7_{\pm0.9}$ | $10.7_{\pm0.8}$ | $42.1_{\pm1.4}$ | $42.2_{\pm4.0}$ | $44.2_{\pm6.4}$ | $\mathbf{57.6}_{\pm4.9}$ |
| Notebook | $10.1_{\pm0.5}$ | $5.9_{\pm0.4}$ | $10.6_{\pm2.9}$ | $31.1_{\pm4.1}$ | $38.1_{\pm4.8}$ | $\mathbf{38.7}_{\pm3.3}$ |
| Scissors | $4.2_{\pm0.5}$ | $4.1_{\pm0.6}$ | $4.4_{\pm0.6}$ | $20.7_{\pm2.0}$ | $35.9_{\pm4.0}$ | $\mathbf{41.4}_{\pm14.9}$ |
| Laptop | $9.9_{\pm0.4}$ | $8.8_{\pm1.1}$ | $33.0_{\pm2.1}$ | $42.5_{\pm5.4}$ | $100_{\pm0.0}$ | $\mathbf{100}_{\pm0.0}$ |

Table 3: Results for the experiments of using one policy per object.

**Baselines.** We compare our approach with the following methods and ablations: (1) *Finger Joint Mapping*: Match the finger joint angles between human and robot hands as demonstration and perform end-to-end imitation learning [43]. (2) *Fingertip Mapping*: Match the fingertip positions between human and robot hands using IK as demonstration and perform end-to-end imitation learning [63]. (3) *Vanilla RL*: Train a PPO policy [70] to learn the motions of the arm and hand together. (4) *Ours*: Full implementation of our hierarchical policy learning approach. (5) *Ours w. FR*: Our method with an additional fingertip-matching reward during training. (6). *Ours w/o. DAL*: Our method without data augmentation loop.

## 5.1 Performance of the high-level planner

**Tasks.** To validate the accuracy of the learned high-level planner, we introduce three experimental tasks: (1) *Trained*: Testing the policy with trained objects and goal trajectories. (2) *Unseen Traj*: Testing the policy conditioned on unseen goal trajectories with trained objects. (3) *Unseen Obj*: Testing the policy with unseen objects and trained goal trajectories.

**Metric.** Evaluation metric is defined as the distance between the ground truth wrist pose trajectory of the hand and the wrist pose trajectory of the left and right hand output by our high-level planner. For each step, we compute translation error (TE) using Euclidean distance in centimeters and orientation error (OE) using angle difference. We report the cumulative error of the entire motion sequence in Table 1

**Results.** Table 1 shows that Ours performs the best in generating wrist motions, with the lowest cumulative translation and orientation error. More importantly, the performance does not drop when testing with unseen goal trajectories (*Unseen Traj*), which demonstrates the generalization capability of our high-level planner to novel object-centric task goals.

## 5.2 Effectiveness of learning from human with hierarchical pipeline

**Task.** To validate the effectiveness of our framework, we first train one policy conditioned on a single trajectory in the training set for each object and test its rollout performance. Each policy is trained on a single object.

**Metric.** We use **Completion Rate** as the evaluation metric, which indicates the percentage of the goal object trajectory completed by the policy. Completion is achieved if and only if the object's pose error is smaller than a threshold (5 centimeters in translation, 2.5 centimeters in object's longest dimension

| | Fingertip Mapping | Finger Joint Mapping | Vanilla RL | Ours (w. FR) | Ours (w.o. DAL) | Ours |
|---|---|---|---|---|---|---|
| Single Obj - Trained Traj | $10.4_{\pm2.8}$ | $7.1_{\pm4.8}$ | $17.1_{\pm9.6}$ | $30.8_{\pm12.1}$ | $59.6_{\pm14.4}$ | $\mathbf{83.8}_{\pm9.1}$ |
| Single Obj - Unseen Traj | $4.8_{\pm0.3}$ | $5.5_{\pm0.8}$ | $18.8_{\pm5.7}$ | $19.6_{\pm10.1}$ | $42.5_{\pm7.5}$ | $\mathbf{57.1}_{\pm10.2}$ |
| Multi Obj - Trained Obj | $6.9_{\pm1.7}$ | $5.3_{\pm1.2}$ | $17.7_{\pm10.1}$ | $15.5_{\pm6.2}$ | $35.2_{\pm2.8}$ | $\mathbf{47.6}_{\pm4.2}$ |
| Multi Obj - Unseen Obj | $3.2_{\pm0.6}$ | $2.9_{\pm0.2}$ | $8.1_{\pm4.2}$ | $8.2_{\pm5.3}$ | $18.6_{\pm3.7}$ | $\mathbf{36.4}_{\pm5.0}$ |

Table 4: Results for the generalization experiments.

multiplied by rotation angle, and $0.5$ radians in joint angle). Each goal trajectory has a total of $500$ action steps.

**Results.** Table 3 demonstrates that our hierarchical learning framework outperforms traditional hand pose matching methods (*Finger Joint Mapping*, *Fingertip Mapping*) by 50% in completion rate, indicating that our low-level RL significantly helps in bridging the embodiment gap when learning from human data. Moreover, *Ours* surpasses *Vanilla RL* by 47.3% on average, underscoring the challenge of training arm and hand actions

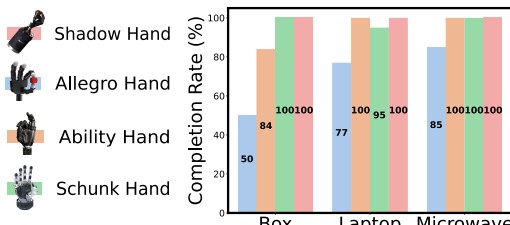

Figure 4: Experiments on the different embodiment. We study four types of the dexterous hand in three tasks from the Section 5.2.

together with RL, and emphasizing the advantage of our high-level planner for guiding the RL in high-dimensional action space. Additionally, the inclusion of human finger motions in the RL reward (*Ours (w. FR)*) does not yield benefits and even leads to lower performance, validating our hypothesis that the embodiment gap makes human finger motion unsuitable for training robot actions. Lastly, our data augmentation loop further brings an additional 7% improvement (*Ours* vs. *Ours (w.o. DAL)*). We further apply our method to four different types of multi-fingered dexterous hands, varying in size and degree of freedom. The results are shown in Figure 4. Our method achieved more than 50% completion rate for all hands, demonstrating that our framework can effectively transfer human data to different robot hand embodiments.

### 5.3 Generalization to unseen scenarios

**Tasks.** We design four types of tasks to test the policy's generalization capability: (1). *Single Obj - Trained Traj*: One policy for each object, and testing with trained goal trajectories. (2). *Single Obj - Unseen Traj*: Same as prior but testing the policy conditioned on unseen goal trajectories. (3). *Multi Obj - Trained Obj*: One policy trained with all objects, and testing with trained objects. (4). *Multi Obj - Unseen Obj*: Same as prior but testing the policy with unseen objects. We use the completion rate as the evaluation metric same as in Section 5.2.

**Results.** In Table 4, our algorithm surpassing the results of *Vanilla RL* on *Single Obj - Unseen Traj* and *Multi Obj - Unseen Obj* by more than 28%. This indicates that our hierarchical structure substantially improves generalization capabilities across unseen trajectories and unseen object geometries. Notably, unlike the Section 5.2, we observe an average 18% improvement in Table 3 with our DAL (*Ours* vs. *Ours (w.o. DAL)*), showcasing that the DAL greatly helps generalization. Traditional mapping methods (*Finger Joint Mapping*, *Fingertip Mapping*) and *Ours (w. FR)* cannot generalize to unseen trajectories and unseen object geometries due to their dependency on the finger information.

### 5.4 Transfer from simulation to real-world

**Tasks.** We train one policy per object conditioned on a single goal trajectory in simulation and test its rollout performance on a real-world robot system. We use the completion rate as the evaluation metric.

**Results.** In Table 2 real-world experiments, our approach has more than a 50% completion rate improvements compared to prior methods, which barely achieve any success ($<20\%$ completion rate) on several objects. This result showcases the ability of our approach on tackling real-world bimaual dexterous manipulation tasks. The visualization of our real-world experiments is shown in Figure 5.

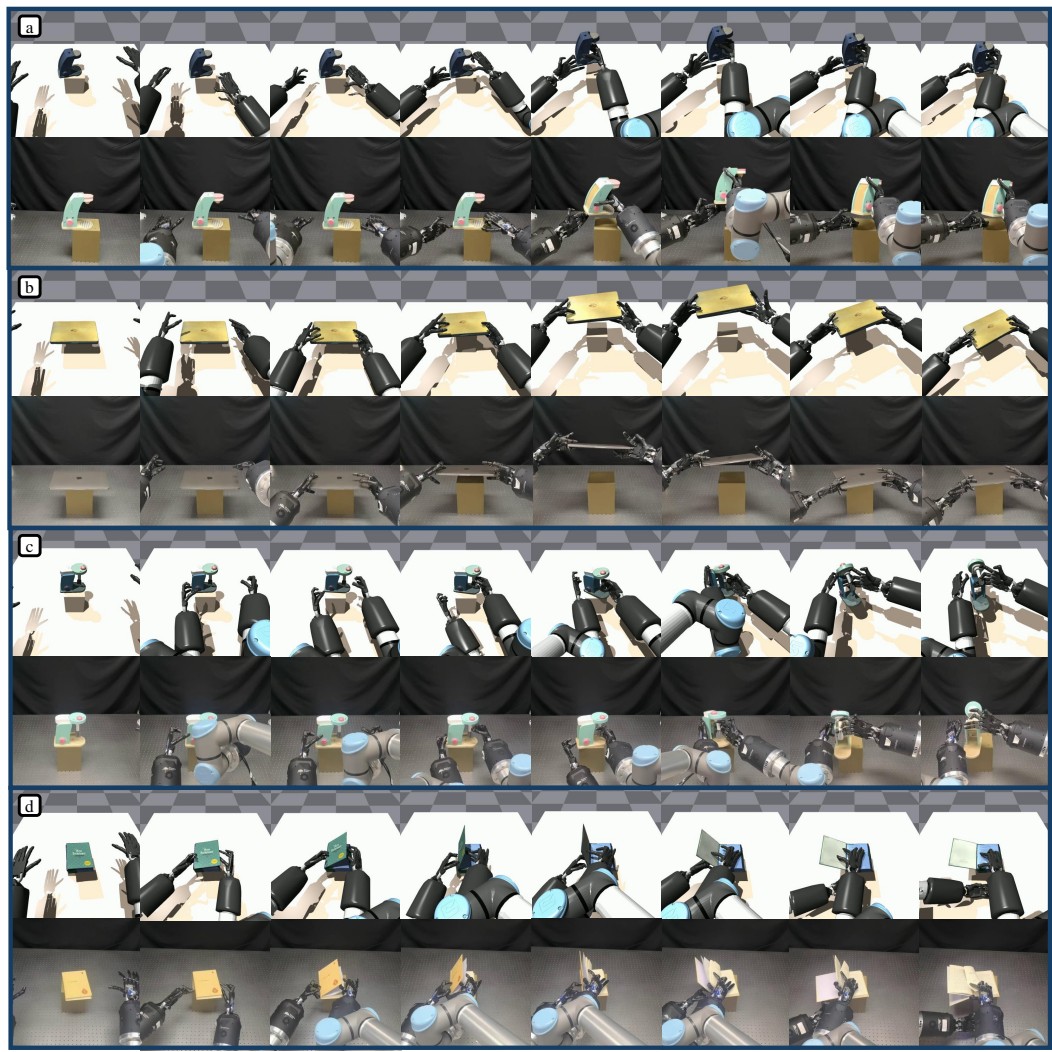

Figure 5: Real-world experiment tasks. All clips include snapshot of the simulation (top row) and the real-world (bottom row). (a) Coffee Maker : Pick and lift the coffee machine. (b) Laptop: Lift up a laptop and place it back to the table. (c) Mixer: Rotate the Mixer and then open it. (d) Notebook: Open the notebook on the table. Please refer to our website for more visualization results.

## 6 Limitations

There are several limitations of our work, including our model encounters difficulties in manipulating small-size objects, the order of joints must be predefined and in our high-level planner, generalizations can only happen in the same category. More discussion about the limitations can be found in Appendix I.

## 7 Conclusion

In this work, we present a hierarchical policy learning framework that effectively utilizes human hand motion data to train object-centric dexterous robot manipulation. At the core of our method is a high-level trajectory generative model trained with a large-scale human hand motion capture dataset, which synthesizes human-like wrist motions conditioned on the object goal trajectory. Guided by these wrist motions, we further trained an RL-based low-level finger controller to achieve the task goal. Our approach demonstrated superior performance across various household objects and showcased generalization capabilities to novel object geometries and goal trajectories. Moreover, the successful transfer of the learned policies from simulation to a real-world bimanual dexterous robot system underscores the practical applicability of our method in real-world scenarios.

**Acknowledgments**

This research was supported by National Science Foundation NSF-FRR-2153854, NSF-NRI-2024247, NSF-CCRI-2120095 and Stanford Institute for Human-Centered Artificial Intelligence, SUHAI.

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

# A  Data Augmentation Loop

## A.1  Pseudo Code

---
**Algorithm 1** DATA AUGMENTATION LOOP

---
**Require:** Human data $D = (G, a^W)$, training set $D_t \in D$, high-level planner $\pi^H$, low-level controller $\pi^L$, augmentation iteration $L$, data augment function $L_{aug}()$, wrist pose trajectories $G_t = (g_t, g_{t+1}, ..., g_{t+T-1})$, goal trajectory of the object and its geometric features $a_t^W = (a_t^W, a_{t+1}^W, ..., a_{t+T-1}^W)$.

  1: Initialize $\pi^H, \pi^L, D_t = \{\}$.
  2: **for** iteration $m = 0,1,...,L$ **do**
  3:     **while** until convergence of $\pi^H$ **do**
  4:         Generate augmented data $L_{aug}(D)$
  5:         Append into training set $D_t \leftarrow D_t + L_{aug}(D)$
  6:         Train $\pi^H$ on $D_t$
  7:     **end while**
  8:     **while** until convergence of $\pi^L$ **do**
  9:         Train $\pi^L$ on $(\pi^H(G), G)$
10:     **end while**
11:     Rollout success trajectories $D_s^t = (a_t^W, G_t)$ with $\pi^L$
12:     Append into human data $D \leftarrow D_T + D_s^t$
13: **end for**

---

## A.2  Detail of the Data Augmentation Loop

Below are the details for each augmentation. The unit of length is centimeters and the unit of angle is degrees.

- Random the object's mesh scales with a small scale:
  - The scale of the width of the manipulated object ranges from 0.9 to 1.1.
  - The scale of the length of the manipulated object ranges from 0.9 to 1.1.
  - The scale of the height of the manipulated object ranges from 0.9 to 1.1.

- Random the object's initial pose with a small scale:
  - The x-coordinate of the manipulated object ranges from -0.02 to 0.02.
  - The y-coordinate of the manipulated object ranges from -0.02 to 0.02.
  - The manipulated object's z-axis Euler degree ranges from 0 to 30.

- Modify the goal trajectories of the object with waypoint interpolation:
  - The x-coordinate of the goal trajectories position is added by ranges from -0.02 to 0.02.
  - The y-coordinate of the goal trajectories position is added by ranges from -0.02 to 0.02.
  - The z-coordinate of the goal trajectories position is added by ranges from -0.02 to 0.02.

# B  Detail Implementation of RL in Simulation

## B.1  Observation Space

Table.5 gives the specific information of the observation space.

## B.2  Reward Design

Denote the $\hat{g}_i^R$, $\hat{g}_i^T$ and $\hat{g}_i^J$ is the current 3D translation, 3D rotation and joint angle of the object respectively, the desired object 3D rotation $g_i^R$, the desired object 3D translation $g_i^T$, and the desired

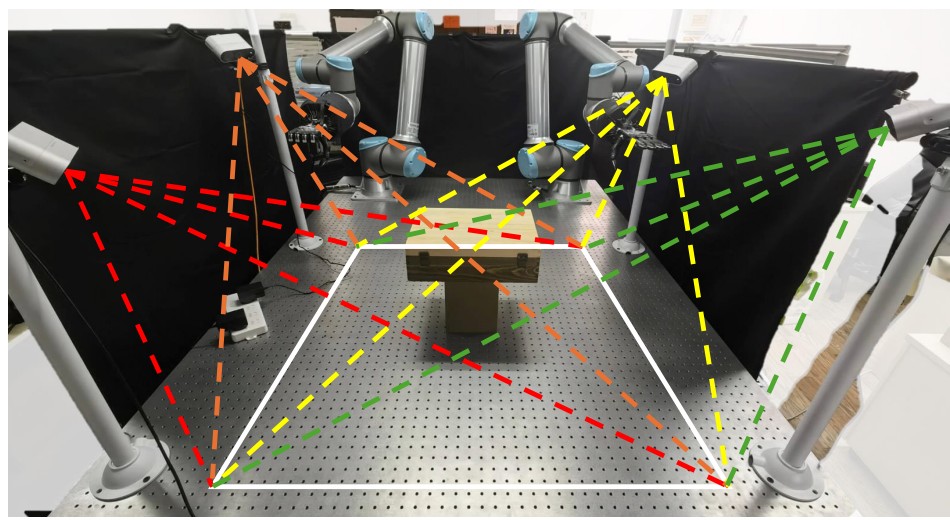

Figure 6: Setup of the cameras.

| Index | Description |
|---|---|
| 0 - 60 | right arm-hand dof position, velocity |
| 60 - 120 | left arm-hand dof position, velocity |
| 120 - 133 | right hand end-effector position, velocity, linear velocity, angle velocity |
| 133 - 146 | left hand end-effector position, velocity, linear velocity, angle velocity |
| 146 - 159 | object base position, rotation, linear velocity, angle velocity |
| 159 - 172 | articulated object top part position, rotation, linear velocity, angle velocity |
| 172 - 185 | articulated object bottom part position, rotation, linear velocity, angle velocity |
| 185 - 187 | object dof position, velocity |
| 187 - 257 | desired object motion trajectory $G = (\boldsymbol{g}_t, \boldsymbol{g}_{t+1}, ..., \boldsymbol{g}_{t+T})$ |
| 257 - 397 | sequence of 6-DoF wrist actions $(\boldsymbol{a}_t^W, \boldsymbol{a}_{t+1}^W, ..., \boldsymbol{a}_{t+T}^W)$ generated by high-level planner |
| 397 - 462 | right hand fingertip pose, linear velocity, angle velocity |
| 462 - 527 | left hand fingertip pose, linear velocity, angle velocity |

Table 5: Observation space of our framework in simulation.

object joint angle $g_i^J$. $\lambda_1$, $\lambda_2$ and $\lambda_3$ is the hyperparameters to balance the weight of each component of the reward.

The reward function is defined as:

$$r_t = \exp^{-(\lambda_1 * \|\boldsymbol{g}_t^R - \hat{\boldsymbol{g}}_t^R\|_2 + \lambda_2 * \|\boldsymbol{g}_t^T - \hat{\boldsymbol{g}}_t^T\|_2 + \lambda_3 * \|g_t^J - \hat{g}_t^J\|_2)} \tag{1}$$

where $\lambda_1 = 20$, $\lambda_2 = 1$, and $\lambda_3 = 5$.

We use an exponential map in the reward function, which is an effective reward shaping technique used in the case to minimize the distance, introduced by [73, 74]. To improve the calculation efficiency, we use quaternion to represent the object orientation. The angular position difference is then computed through the dot product between the normalized goal quaternion and the current object's quaternion.

## C   Detail Implementation in Real-World

### C.1   Perception

Our perception setup is shown in Figure 6. We arranged 4 identical Femto Bolt cameras around the table and face towards the object. We use FoundationPose [72] to estimate the articulated object pose. To remove the abnormal results, we compare each pose to the desired pose and remove the pose if the

error is smaller than a threshold (5 centimeters in translation and $0.5$ radians in orientation). Finally, we average the rest of the poses as our observation for the policy. If none of the poses is smaller than the threshold, we continue to use the pose from the previous frame.

## C.2 Policy Distillation

We use the DAgger [75] algorithm for policy distillation. Table.6 gives the specific information of the observation space of the distilled policy.

| Index | Description |
|---|---|
| 0 - 24 | right hand dof position |
| 24 - 48 | left hand dof position |
| 48 - 55 | right hand end-effector position, rotation |
| 55 - 62 | left hand end-effector position, rotation |
| 62 - 69 | articulated object top part position, rotation |
| 69 - 76 | articulated object bottom part position, rotation |
| 76 - 77 | object dof position |
| 77 - 147 | desired object motion trajectory $G = (\boldsymbol{g}_t, \boldsymbol{g}_{t+1}, ..., \boldsymbol{g}_{t+T})$ |
| 147 - 287 | sequence of 6-DoF wrist actions $(\boldsymbol{a}_t^W, \boldsymbol{a}_{t+1}^W, ..., \boldsymbol{a}_{t+T}^W)$ generated by high-level planner |

Table 6: Observation space of our framework in the real-world.

## C.3 Resets

In the real-world evaluation, we used a trajectory from the ARCTIC dataset as our goal trajectory. In terms of resets in our real-world experiments, we pose the object within 3 centimeters and $0.5$ radians of the initial pose of the goal trajectory during resets. We evaluated 20 times and reported the completion rate.

## C.4 Evaluation

When evaluating real-world experiments, we take the same metric (completion rate) as in simulation. It will fail if the tolerance is exceeded(5 centimeters in translation, $2.5$ centimeters in object's longest dimension multiplied by rotation angle, and $0.5$ radians in joint angle), and record the completion rate.

## D Hyperparameters of the PPO

Table.7 gives the hyperparameters of the PPO.

| Hyperparameters | Value |
|---|---|
| Num mini-batches | 4 |
| Num opt-epochs | 5 |
| Num episode-length | 8 |
| Hidden size | [1024, 1024, 512, 256] |
| Clip range | 0.2 |
| Max grad norm | 1 |
| Learning rate | 3.e-4 |
| Discount ($\gamma$) | 0.998 |
| GAE lambda ($\lambda$) | 0.95 |
| Init noise std | 0.8 |
| Desired kl | 0.02 |
| Ent-coef | 0 |

Table 7: Hyperparameters of PPO.

# E  Domain Randomization

Isaac Gym provides lots of domain randomization functions for RL training. We add the randomization for all the tasks as shown in Table. 8 for each environment. we generate new randomization every 1000 simulation steps.

| Parameter | Type | Distribution | Initial Range |
|---|---|---|---|
| **Robot** | | | |
| Mass | Scaling | uniform | [0.5, 1.5] |
| Friction | Scaling | uniform | [0.7, 1.3] |
| Joint Lower Limit | Scaling | loguniform | [0.0, 0.01] |
| Joint Upper Limit | Scaling | loguniform | [0.0, 0.01] |
| Joint Stiffness | Scaling | loguniform | [0.0, 0.01] |
| Joint Damping | Scaling | loguniform | [0.0, 0.01] |
| **Object** | | | |
| Mass | Scaling | uniform | [0.5, 1.5] |
| Friction | Scaling | uniform | [0.5, 1.5] |
| Scale | Scaling | uniform | [0.95, 1.05] |
| Position Noise | Additive | gaussian | [0.0, 0.02] |
| Rotation Noise | Additive | gaussian | [0.0, 0.2] |
| **Observation** | | | |
| Obs Correlated. Noise | Additive | gaussian | [0.0, 0.001] |
| Obs Uncorrelated. Noise | Additive | gaussian | [0.0, 0.002] |
| **Action** | | | |
| Action Correlated Noise | Additive | gaussian | [0.0, 0.015] |
| Action Uncorrelated Noise | Additive | gaussian | [0.0, 0.05] |
| **Environment** | | | |
| Gravity | Additive | normal | [0, 0.4] |

Table 8: Domain randomization of all the tasks.

# F  Goal Representation

Our framework requires the full 6D pose of the object trajectory at each time-step. For downstream tasks, we are able to get the trajectories in many ways. For example, we can specify a few key poses of the object based on the task, and use linear interpolation to generate the pose trajectories between the key poses. Another solution is to use some object motion synthesis methods in the field of graphics to generate the 6D pose of the object trajectory, such as [76]. We design an additional experiment to prove that our method can work with this setup. Taking a task "lifting up the box in the air and dropping it down" as an example, we can just simply manually define the pose of the box in the air and the landing point after picking it up as the key poses, then interpolate it as our goal trajectory, and train the robot to manipulate the object to follow the goal trajectory. The results are shown in Table 9 and the snapshot are shown in Figure 7. The results show that we can complete the task under this setting, and shows that our framework is extensible.

| | Box | Manually Designed Trajectory |
|---|---|---|
| Our | $100_{\pm 0.0}$ | $100_{\pm 0.0}$ |

Table 9: Results for the goal representation experiments.

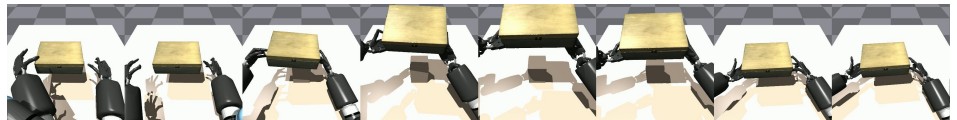

Figure 7: Snapshots of the goal representation experiment.

## G  Time-indexed of the Goal Trajectory

For the time-indexed of the goal trajectory, we randomize the sampling of input trajectories with various time gaps during training. We added an additional experiment to show that varying the time gap does not significantly affect our performance, indicating that our approach maintains generalization despite these temporal constraints. Using Coffee Maker as an example, we interpolated the goal trajectories provided by ARCTIC datasets to varying levels, and tested the completion rate of our policy on it. Origin represents the original goal trajectory from ARCTIC, and Skip 2 and Skip 1 represent skipping 2/1 poses between every two poses on the basis of origin goal trajectory. Inter 2 and Inter 1 represent insertion of 2/1 pose between every two poses according to the linear interpolation method on the basis of origin goal trajectory. Our policy generates robot actions conditioned on a sequence of object goal poses, with the time gaps defined by the distance between each consecutive goal pose. In this experiment, we randomly use time gaps of 0 to 3 units when sampling the object trajectory to test whether the policy is sensitive to these time gaps (Random Sample). The results is shown in Table 10, the performance of the trained policy will not vary greatly.

|     | Skip 2 | Skip 1 | Origin | Inter 1 | Inter 2 | Random Sample |
|-----|--------|--------|--------|---------|---------|---------------|
| Our | $85.7_{\pm 7.5}$ | $86.5_{\pm 3.4}$ | $86.1_{\pm 5.5}$ | $85.4_{\pm 8.1}$ | $87.2_{\pm 4.8}$ | $86.2_{\pm 6.2}$ |

Table 10: Results for the time-indexed experiments.

## H  Model-base Baselines

We add a baseline of the model-based method in the simulation, the result is shown in Table 11. We use the sample-based model predictive control method that is modified from [77] as our low-level controller in our tasks. Our method outperforms the sample-based model predictive control by 46.3% on average.

## I  Limitation

There are several limitations of our work. Firstly, our model encounters difficulties in manipulating small-size objects (e.g. scissors, ketchup, etc.). One potential direction is to make better use of contact information in human data. For example, [69] uses a contact graph in reward function to assist the reinforcement learning, which has the possibility of being incorporated into our framework. Secondly, we did not leverage the tactile information in the ARCTIC dataset during policy learning. In future work, we plan to equip our robot hands with tactile sensors and use contact information to train the robot to handle more contact-rich manipulation tasks. Thirdly, our method sometimes fails when the difference between the initial pose of the real-world and the goal trajectory is large. Another failure is due to the dexterous hand forms a large occlusion of the object, resulting in inaccurate pose estimation in the real-world. Fourth, The order of joints must be predefined. This limitation may not pose significant issues with single revolute joints. However, complications arise when objects feature multiple revolute joints, as the representation lacks permutation invariance across joint orders. We addressed this by focusing on category-level joint inputs consistent with URDF definitions, leveraging datasets like ShapeNet that maintain joint order consistency within categories. Future steps will involve exploring robust methods to accommodate diverse joint configurations beyond current dataset limitations. Fifth,

| | Box | Coffee Maker | Espresso Machine | Ketch | Micro-wave | Mixer | Note-book | Scis-sors | Laptop |
|---|---|---|---|---|---|---|---|---|---|
| MPC | $31.6_{\pm1.5}$ | $22.8_{\pm0.7}$ | $30.4_{\pm2.3}$ | $24.8_{\pm1.3}$ | $28.6_{\pm1.9}$ | $26.8_{\pm0.5}$ | $19.2_{\pm2.0}$ | $23.6_{\pm0.8}$ | $24.0_{\pm1.4}$ |
| Ours | $\mathbf{100}_{\pm0.0}$ | $\mathbf{86.1}_{\pm5.5}$ | $\mathbf{81.1}_{\pm8.6}$ | $\mathbf{41.2}_{\pm13.3}$ | $\mathbf{100}_{\pm0.0}$ | $\mathbf{57.6}_{\pm4.9}$ | $\mathbf{38.7}_{\pm3.3}$ | $\mathbf{41.4}_{\pm14.9}$ | $\mathbf{100}_{\pm0.0}$ |

Table 11: Results for the experiments of using one policy per object.

in our high-level planner, generalizations can only happen in the same category. But we still can force the model to predict action for unseen categories by sending in a trained category id and test with unseen categories. This is how we did for the generalization experiments by using the trained Box policy to manipulate the Laptop. Since there are only 11 objects in the ARCTIC dataset, it is difficult to test the capability of the generalizations across categories. In future works we will use larger 3D object dataset and human motion capture dataset to test the generalization to unseen categories.

