# OpenReview forum: "Object-Centric Dexterous Manipulation from Human Motion Data"
_robot-learning.org/CoRL/2024/Conference — CoRL 2024_

### Official Review · Reviewer_u9EE · 2024-07-02
**Great results on a challenging control task**

**Originality:** 3
**Technical Quality:** 4
**Clarity Of Presentation:** 3
**Potential Impact:** 3
**Recommendation:** 4
**Confidence:** 3

**Review:**

**Update (after rebuttal)**

During the rebuttal period, the authors improved their explanations of the Experiments section, and clarified some missing details. With these changes, I'm confident the paper should be accepted to CoRL.

**Original review**

Quality:

This paper presents some excellent results in simulation, with decent transfer to real, on complicated manipulation tasks.
While I have confidence that the system works (based on the shared videos), the Experiments section lacks a lot of detail and is hard to understand.

Clarity:

With the exception of the Experiments section (expanded below), the paper is well-written and clear. The method is motivated and described well.

Originality:

The idea of separating large arm movements from fine low level control has been tried before, as well as the idea of imitating human data for the large arm movements.
The most similar work I could find was (Sudeep Dasari; Abhinav Gupta; Vikash Kumar, 2023), where human data was used to generate a pre-grasp position for the arm, and RL was used to control the arm and fingers from then on.
This paper introduces a couple of advances (bimanual manipulation, real robot experiments, full use of recorded wrist trajectories instead of just a pre-grasp) that make it sufficiently interesting.

Significance:

This work shows successful results on real-robot manipulation of articulated bodies using bimanual multifingered robots, and will certainly inspire future works in this domain.

Strengths:

The problem tackled is challenging, and the approach is explained clearly. The real robot results are impressive.

Weaknesses:

The Experiments section is hard to parse and lacks a lot of details, to explain what numbers are being presented. While reading it, I came up with the following questions:

* In section 5.1, when evaluating different high level planners, is the low-level controller frozen or trained separately for each high level planner?

* In table 1, orientation errors are reported as Frobenius norm, which is hard to conceptualize. Since in other parts of the paper you talk about angle difference (e.g the completion criterion on line 200), could you convert those numbers to angle difference too?

* In section 5.2, the definition of the completion rate seems very lenient. In particular, a 0.5 radian (28 degree) error seems impractical for any application, particularly for large objects. Given a few of the results saturate at 100%, reducing this threshold seems sensible. Either choosing a small number (e.g. 0.1 radian), or scaling it by the object's longest dimension (e.g. L * angle < 2.5cm) would be fine.

* In section 5.2, why doesn't the completion criterion include a mention of the articulation of the object?

* I couldn't find out what the $\pm$ figures in table 3 are. Are they confidence intervals or standard deviations? Why do some results with similar success probabilities have very large differences in the confidence interval? Were some evaluations run more times than others? Please include this information in the appendix. For estimating confidence intervals for a binomial proportion, consider using something other than the normal approximation (https://en.wikipedia.org/wiki/Binomial_proportion_confidence_interval).

* Table 2 shows success rates at three significant figures for real world evaluations, but I wasn't able to find the number of executed trajectories in those evaluations. Could this data be included in the appendix? Probably, the number of significant figures should be reduced. How was evaluation done in the real world (in terms of resets, success measurement)?

* In the supplementary video, some sim policies have significant vibrations in the arms that aren't present on the real robot. Is this a result of sim instability or of the low-pass filter mentioned in section 4.4?

**Quality Of The Limitations Section:**

3

**Questions For Rebuttal:**

* Please provide all the details required to understand the numbers presented in the experimental evaluations.
* When referring to (Sudeep Dasari; Abhinav Gupta; Vikash Kumar, 2023), please explicitly address the differences between your approach and theirs, as they are similar (and got similar sim results).
* Please add a citation to an early work showing independent control of the fingers and arm, e.g. http://yutingye.info/SIG12.html.

**Robotics Focus:**

4

**Summary Of Paper:**

The authors tackle the problem of manipulating objects using a bimanual fingered robot by decomposing it into two parts: generating rough wrist positions, and generating fine finger motions. The first subproblem is solved by imitating human-recorded trajectories, while the second is solved by training policies using RL, conditioned on the goal trajectory of the object, and on the wrist trajectory generated by the imitating controller. The method is applied to the challenging problem of tracking trajectories of articulated objects through space, using two five-fingered robots. They transfer the learned behaviours from sim to real, using a mix of explicit object state estimation and teacher-student distillation.

**Summary Of Recommendation:**

This is a strong paper with impressive performance on a real complex embodiment.

---

### Official Review · Reviewer_2mos · 2024-07-13
**Hierarchical learning framework shows promising generalization**

**Originality:** 3
**Technical Quality:** 3
**Clarity Of Presentation:** 4
**Potential Impact:** 3
**Recommendation:** 3
**Confidence:** 4

**Review:**

Strength:
- The paper is well written and easy to follow.
- The paper performs relatively extensive evaluations in both simulation and hardware to validate their hierarchical learning framework.
- The demonstration of sim-to-real transfer is relatively convincing.

Weakness:
- Lack of discussion on model-based techniques: The paper didn’t mention other relevant dexterous manipulation works using model-based optimization techniques such as “Global Planning for Contact-Rich Manipulation via Local Smoothing of Quasi-Dynamic Contact Models” and “Enhancing dexterity in robotic manipulation via hierarchical contact exploration” etc.
- Comparison with model-based methods: The paper focuses on comparisons with other learning-based approaches but doesn't compare against model-based optimization techniques, which are relevant in the field of dexterous manipulation.
- Failure analysis: There's limited discussion on failure modes or edge cases where the method underperforms, which could provide valuable insights for future improvements.

**Quality Of The Limitations Section:**

3

**Questions For Rebuttal:**

1. Why don’t Table 1 and 2 have variances?
2. It would be great to clarify if the policy for Table 3 is trained for single or multiple objects.
3. Could the authors provide some details about why allegro hand has much lower completion rate compared to the other 3 hands?

**Robotics Focus:**

4

**Summary Of Paper:**

The paper proposes a hierarchical policy learning framework that learns high-level wrist movements from human motion with IL and lower-level finger movements with RL. The experimental results show promising generalization in objects and trajectories.

**Summary Of Recommendation:**

The core idea of leveraging human motion data to guide robot manipulation is clear and supported by substantial experimental results.

---

### Official Review · Reviewer_vhQ2 · 2024-07-21
**Interesting Formulations and Good Results, Need to Clarify Several Details**

**Originality:** 3
**Technical Quality:** 3
**Clarity Of Presentation:** 3
**Potential Impact:** 3
**Recommendation:** 3
**Confidence:** 4

**Review:**

Strength:

The formulation of defining tasks according to object trajectory, while ignoring the wrist/finger trajectories, is reasonable given the embodiment gap. I believe this is an important step forward in learning more diverse tasks using sim-to-real.

Separating the learning of wrist and finger movements is interesting and shows significant performance gains. It also shows the advantage of being transferable to different hands (though only in simulation).

The demonstration of sim-to-real bimanual dexterous manipulation is impressive.

Weakness:

One limitation of this paper is the goal representation. It requires the full 6D pose of the object trajectory at each timestep, which is a strict requirement for downstream applications, as many tasks do not have such data. The representation also needs to be time-indexed, meaning that the robot needs to follow specific object velocities, and the timestep must be tuned when specifying the goal. This will limit the generalization of the formulation.

The order of joints needs to be fixed, implying that the simulation and reality need to be closely matched. This might not be problematic when there is only one revolute joint. However, it becomes problematic when objects have more than one revolute joint, in which case the representation is not permutation invariant over the joint order.

How many trials are used in the evaluation? The main text does not provide this information, and the website only displays three, which is a very small number of trials.

In the high-level planner, the policy takes the object category id c as input. It is unclear how this can generalize to different categories. Or does the generalization only happens for unseen objects within the same category?

**Quality Of The Limitations Section:**

2

**Questions For Rebuttal:**

When given the goal object trajectory as input, does the policy have a certain level of robustness for noisy goals? For example, if one adding position/orientation noise to the trajectory, will the policy still perform reasonably well?

How many trials are used in the real and simulation evaluations, respectively?

Does it generalize to unseen categories or unseen objects with the same trajectory? If it only generalize within the same category, this should be discussed in the limitation section.

For the high-level planner, what is the actual time (in terms of seconds) corresponding to timestep=10?

What are the difference between the proposed method and BC / BC-RNN in Table 1? I think both of them are supervised learning so I’m not sure what the authors want to show here?

**Robotics Focus:**

4

**Summary Of Paper:**

This paper proposes a method to learn dexterous manipulation specified by goal object trajectories. It divides the task into two stages: First, a policy generates hand wrist trajectories from given object trajectories, learned from human videos. Second, using the wrist trajectory, it uses RL to learn the finger motions. Experiments demonstrate that the proposed framework outperforms both vanilla RL and single-stage policies.

**Summary Of Recommendation:**

This paper presents good results with interesting formulations. I recommend acceptance but also emphasize that many details should be further clarified.

---

### Author Rebuttal · Authors · 2024-08-08

Dear Reviewers and Meta-Reviewer,

We sincerely appreciate the time you dedicated to providing us with constructive feedback and valuable advice to enhance our paper. In response, we have revised the manuscript and conducted additional experiments to offer deeper insights and address your concerns. In the responses to each reviewer below, we have addressed your specific questions and comments. The suggested revisions have been implemented and are highlighted in red throughout the paper. We look forward to any further discussions!

[Updated on 08-13: Paper revised]

---

### Decision · Program_Chairs · 2024-09-04

**Decision:**

Accept

**Comment:**

**Pre-rebuttal**

This paper receives three positive reviews. The reviewers acknowledge the soundness of the idea and its promise for learning bimanual manipulation from human motion data. Other mentioned merits include the novelty of the method (**u9EE**) and the clarity of writing (**u9EE**, **2mos**). Particularly, all three reviewers (**vhQ2**, **u9EE**, **2mos**) acknowledge the impressiveness of the sim-to-real demonstration.

The main mentioned weaknesses include:
- The assumption of dense object 6D pose in the demonstration and the rigidity of object representation (**vhQ2**).
- Unclear extent of generalization (**vhQ2**).
- Missing details and unclear presentation in the experiments section (**vhQ2**, **u9EE**).
- Comparison with model-based methods (**2mos**).
- Failure analysis and discussion (**2mos**).

---
**Post-rebuttal**

This paper initially received three weak accepts. All the reviewers remain positive after the rebuttal. Following the rebuttal iteration, one reviewer (**u9EE**) raised the assessment to strong accept.

Overall the paper has delivered an impressive sim-to-real pipeline for bimanual dexterous manipulation from human mocap data. AC agrees with the assessments of the reviewers and thereby recommends accept.

Other feedback:
- After reading all the reviews/rebuttal, AC had the same question as **u9EE** of *"Table 2 shows success rates at three significant figures ... How was evaluation done in the real world (in terms of resets, success measurement)?"*, and didn't find the rebuttal response effective. "20 times of evaluation" will produce an increment interval of 5% in the completion rate and does not explain for the three significant figures here. Assuming evaluating on multiple seeds, one would need 50 seeds to produce an increment interval of 0.1% as given in Tab. 2. AC encourages the authors to further clarify this in the follow-up revision.

Typos:
- [Line 190] "Table" -> "Table."